

# Cloning, expression and characterization of a chitinase from *Paenibacillus chitinolyticus* strain UMBR 0002

Cong Liu[1], Naikun Shen[2], Jiafa Wu[2], Mingguo Jiang[2], Songbiao Shi[1], Jinzi Wang[2], Yanye Wei[2] and Lifang Yang[1]

[1] Guangxi Key Laboratory of Chemistry and Engineering of Forest Products, Guangxi Key Laboratory for Polysaccharide Materials and Modifications, School of Chemistry and Chemical Engineering, Guangxi University for Nationalities, Nanning, China
[2] School of Marine Sciences and Biotechnology, Guangxi University for Nationalities, Nanning, China

Corresponding authors
Mingguo Jiang,
mzxyjiang@gxun.edu.cn
Lifang Yang,
yanglf1990@gxun.edu.cn

## ABSTRACT

**Background:** Chitinases are enzymes which degrade β-1,4-glycosidid linkages in chitin. The enzymatic degradation of shellfish waste (containing chitin) to chitooligosaccharides is used in industrial applications to generate high-value-added products from such waste. However, chitinases are currently produced with low efficiency and poor tolerance, limiting the industrial utility. Therefore, identifying chitinases with higher enzymatic activity and tolerance is of great importance.
**Methods:** Primers were designed using the genomic database of *Paenibacillus chitinolyticus* NBRC 15660. An exochitinase (CHI) was cloned into the recombinant plasmid pET-22b (+) to form pET-22b (+)-CHI, which was transformed into *Escherichia coli* TOP10 to construct a genomic library. Transformation was confirmed by colony-polymerase chain reaction and electrophoresis. The target sequence was verified by sequencing. Recombinant pET-22b (+)-CHI was transformed into *E. coli* Rosetta-gami B (DE3) for expression of chitinase. Recombinant protein was purified by Ni-NTA affinity chromatography and enzymatic analysis was carried out.
**Results:** The exochitinase CHI from *P. chitinolyticus* strain UMBR 0002 was successfully cloned and heterologously expressed in *E. coli* Rosetta-gami B (DE3). Purification yielded a 13.36-fold enrichment and recovery yield of 72.20%. The purified enzyme had a specific activity of 750.64 mU mg$^{-1}$. The optimum pH and temperature for degradation of colloidal chitin were 5.0 and 45 °C, respectively. The enzyme showed high stability, retaining >70% activity at pH 4.0–10.0 and 25–45 °C (maximum of 90 min). The activity of CHI strongly increased with the addition of Ca$^{2+}$, Mn$^{2+}$, Tween 80 and urea. Conversely, Cu$^{2+}$, Fe$^{3+}$, acetic acid, isoamyl alcohol, sodium dodecyl sulfate and β-mercaptoethanol significantly inhibited enzyme activity. The oligosaccharides produced by CHI from colloidal chitin exhibited a degree of polymerization, forming N-acetylglucosamine (GlcNAc) and (GlcNAc)$_2$ as products.
**Conclusions:** This is the first report of the cloning, heterologous expression and purification of a chitinase from *P. chitinolyticus* strain UMBR 0002. The results highlight CHI as a good candidate enzyme for green degradation of chitinous waste.

## INTRODUCTION

Chitin is composed of linear chains of poly-β-1,4-N-acetylglucosamine, and is found in the exoskeletons of arthropods and fungal cell walls. Chitin is the second most abundant and renewable natural polysaccharide after cellulose (*Li et al., 2010*; *Yang et al., 2009*; *Karthik, Binod & Pandey, 2015*), and forms the structural components of the cells of numerous organisms (*Rameshthangam et al., 2018*). However, chitin waste does not degrade readily; an attractive solution would be to extract chitin and its derivatives from chitin-rich waste and convert the waste into high-value-added products (*Nawani et al., 2002*). The degradation products of chitin have many properties including antifungal activity, biocompatibility, biodegradability, non-toxicity and adsorption, meaning that they have utility in many fields, such as materials and medical sciences, food and nutrition, biotechnology, agriculture and environmental protection (*Guo et al., 2019*; *Hamed, Özogul & Regenstein, 2016*; *Meena et al., 2014*). Chitinases (EC 3.2.2.14) catalyze the degradation of chitin by cleaving the β-1,4 linkages of N-acetylglucosamine units (*Salas-Ovilla et al., 2019*). Bioprocesses for recycling and degrading waste chitin using chitinase have been adopted for the industrial production of GlcNAc (*Das et al., 2019*). At present, degradation products of chitin are usually generated through chemical degradation, which not only has high energy consumption but also creates serious environmental pollution. However, degradation of chitin by chitinases is controllable and low cost and can produce high yields in mild conditions with minimal environmental impacts (*Deng et al., 2019*; *Kidibule et al., 2018*; *Salas-Ovilla et al., 2019*). In nature, more than $10^9$ tons of chitin is produced each year (*Menghiu et al., 2019*), and over 10,000 tons of shellfish waste is produced annually, most of which is thrown into the sea or land fill. If this waste is not fully decomposed, it may have negative effects on the environment and human health (*Rameshthangam et al., 2018*; *Wang et al., 2018*). Reducing the large amount of waste and creating high-value-added products from chitin waste is of high importance.

Chitinases can be subdivided into three families based on their structural motives: the glycoside hydrolase families 18, 19 and 20 (*Take et al., 2018*). Chitinases can be further classified as endochitinases, exochitinases, or N-acetylglucosaminases (NAGases) based on the mechanism of chitin hydrolysis. Endochitinases (EC 3.2.1.14) randomly cleave chitin at internal sites to produce soluble, low-molecular-mass GlcNAc oligomers. Exo-chitinases can be subdivided into chitobiosidases (EC 3.2.1.29) and 1-4-β-glucosaminidases (EC 3.2.1.30). The former releases (GlcNAc)$_2$ from the non-reducing end of chitin chains. The latter acts on β-1, 4-glycosidic bonds at the end of products of low-molecular-mass GlcNAc oligomers and chitobiosidases to generate GlcNAc monomers (*Hamid et al., 2013*; *Zhou et al., 2019*). Bacterial chitinase offers wide environmental adaptability, fast expression, high thermal stability; furthermore, the enzyme can be genetically engineered (*Le & Yang, 2019*). Chitinase-producing bacteria

have been isolated from many sources such as soil, shellfish waste, compost and hot springs (*Hamid et al., 2013*). Many bacteria can produce chitinases, including *Streptomyces*, *Pseudomonas*, *Sanguibacter*, *Paenibacillus*, *Serratia*, *Fusarium*, *Clostridium*, *Flavobacterium*, *Alteromonas*, *Bacillus*, *Microbispora*, *Chromobacterium*, *Erwinia* and *Vibrio* (*Bouacem et al., 2018*; *Das et al., 2019*; *Nawani et al., 2002*; *Stoykov, Pavlov & Krastanov, 2015*; *Vaikuntapu et al., 2016*). Many of these produce chitinases which work at moderate temperature or neutral conditions (*Bouacem et al., 2018*). However, chitinases are usually tolerant to, for example, extremes of pH and temperature for long periods, making them suitable for use in practical industrial applications (*Karthik, Binod & Pandey, 2015*). Compared with those derived from terrestrial organisms, marine chitinases usually exhibit higher tolerance of pH and salinity (*Das et al., 2019*; *Han et al., 2009*). Therefore, it is important to screen for highly tolerant, novel marine chitinases.

In our previous study, we isolated a marine bacterium capable of producing chitinase from a sample of shrimp-pond bottom mud from mangroves in the Maowei Sea in China, which we identified to be *Paenibacillus chitinolyticus* strain UMBR 0002. The majority of previous studies on *P. chitinolyticus* have focused on the wild bacterium (*Song et al., 2012*; *Mihajlovski et al., 2017*). As far as is known, few studies have been reported that chitinase genes from *P. chitinolyticus* have been cloned. In the present study, we report the cloning, heterologous expression and purification of a chitinase (CHI) from *P. chitinolyticus* strain UMBR 0002 for the first time. The enzymatic properties and products were analyzed to provide a theoretical basis for effective utilization of this enzyme for the degradation of chitin waste.

## METHODS

### Materials

We obtained DNA polymerases, *Bam*HI and *Xho*I restriction endonucleases and $T_4$ ligase from TaKaRa (Dalian, China). The TIANamp Bacteria DNA Kit, TIANprep Mini Plasmid Kit, and Universal DNA Purification Kit were all purchased from Tiangen Biotech CO., Ltd. (Beijing, China). EasyPure Quick Gel Extraction Kit was purchased from TransGen Biotech CO., Ltd. (Beijing, China). His-Tagged Protein Purification Kit (Soluble Protein) was obtained from ComWin Biotech Co., Ltd. (Beijing, China). Host cell strains, *E. coli* TOP10 and Rosetta-gami B (DE3), were obtained from TransGen Biotech CO., Ltd. and Sangon Biotech Co., Ltd. (Shanghai, China) and the pET-22b (+) was purchased from Sangon Biotech Co., Ltd. Chitin powder (from crab shells), chitosan oligosaccharide degree of deacetylation (DDA) >90%, carboxymethyl cellulose sodium salt (CMC-Na), and carboxymethyl chitosan (DDA ≥ 90%) were obtained from Solarbio (Beijing, China), Dalian Zhongke Glake Biotechnology Co., Ltd. (Dalian, China), Damao Chemical Reagent Factory (Tianjin, China) and Hefei Bomei Biotechnology Co., Ltd. (Hefei, China), respectively. All other chemicals used in this study were of analytical grade. The *P. chitinolyticus* strain UMBR 0002 was reserved in the Guangdong culture collection center under the number GDMCC 60710.

## Strains and culture conditions

*Paenibacillus chitinolyticus* strain UMBR 0002 was grown in sterile chitin medium at pH 7.0 containing (in g $L^{-1}$): colloidal chitin 3.0, tryptone 5.0, inorganic salt mixture 10.0 and agar powder 15.0 (for solid culture medium). The inorganic salt mixture contained (in g $L^{-1}$): $KNO_3$ 100.0, $K_2HPO_4$ 50.0, $NH_4NO_3$ 10.0, $MgSO_4 \cdot 7H_2O$ 50.0, NaCl 50.0, $FeSO_4$ 1.0, $MnCl_2 \cdot H_2O$ 0.1 and $ZnSO_4 \cdot 7H_2O$ 0.1.

The *E. coli* TOP10 and Rosetta-gami B (DE3) strains were cultivated in sterile Luria-Bertani (LB) medium at pH 7.4 ± 0.2, containing (in g $L^{-1}$): tryptone 10.0, yeast extract 5.0, NaCl 10.0, and agar powder 15.0 (for solid culture medium).

## Preparation of colloidal chitin

Colloidal chitin was prepared according to the method of *Laribi-Habchi et al. (2015)* with minor modifications. Briefly, 10.0 g of chitin powder dissolved in 176 mL of concentrated hydrochloric acid (12 M), stirred well and stored at 4 °C for 24 h. Thereafter, 1.0 L of distilled water was added to the chitin solution and the mixture incubated for another 24 h at 4 °C. The precipitate was harvested by centrifugation at 8,000 rpm for 10 min and washed repeatedly with sterile distilled water until the pH reached 7.0. Finally, the precipitate was dissolved in sterile distilled water to a final concentration of 25.0 g $L^{-1}$. The solution was stored at 4 °C for subsequent use.

## Activity assay

The specific activity (mU $mg^{-1}$ protein) of purified recombinant CHI and crude protein extracts solution toward colloidal chitin were determined according to the method described by *Miller (1959)*. Diluted recombinant protein solution (100 µL) in phosphate buffer was mixed with colloidal chitin (at a final concentration of 0.5%), and incubated for 1 h at 45 °C along with appropriate blanks. The reaction was stopped by the addition of 300 µL 3,5-dinitrosalicylic acid (DNS) followed by heating for 10 min in a boiling water bath. After the reaction system was cooled to room temperature, 300 µL of deionized water was added to the mixture, and the resulting solution was centrifuged at 8,000 rpm for 10 min. The optical density (OD) of the supernatant was measured at 540 nm using a BioTek spectrophotometer (Epoch, Winooski, VT, USA). Chitinase activity was determined by colorimetry, based on the amount of GlcNAc released from colloidal chitin. This was calculated by comparing $OD_{540}$ data with a standard curve prepared from serial dilutions of GlcNAc (from 0.1 to 1 mg $mL^{-1}$) (*Mohamed et al., 2019*). One unit of chitinase activity was defined as the amount of enzyme that releases 1 µmoL of GlcNAc from colloidal chitin per min under the specified assay conditions.

## Plasmids and cloning of the chitinase gene

Oligonucleotide primers used for the cloning of the chitinase (CHI) gene from *P. chitinolyticus* strain UMBR 0002 were designed from the genomic database of *P. chitinolyticus* NBRC 15660 (GenBank accession number NZ_BBJT00000000.1). The nucleotide sequence of CHI gene was deposited in GenBank with the accession number MN121846. A DNA extraction kit was used to obtain DNA, which was then used as the template for

polymerase chain reaction (PCR) amplification. The gene encoding CHI was amplified using *Pfu* DNA polymerase and the following primer pairs: 5′-GGATCCCGAACC GGCCAAAATCGTCGG-3′ and 5′-CTCGAGGCTGCCTGTTACCACAATATTCG-3′ (underlining indicates the added *Bam*HI and *Xho*I sites). The PCR program was as follows: initial denaturation at 98 °C for 8 min, followed by 30 cycles of 98 °C for 30 s, 55 °C for 45 s and 72 °C for 2 min 30 s, with a final extension step at 72 °C for 10 min. The PCR samples were purified using a Universal DNA Purification Kit, digested with *Bam*HI and *Xho*I and then ligated into a *Bam*HI- and *Xho*I-digested vector pET-22b (+) to generate pET-22b (+)-CHI. The recombinant plasmid was transformed into *E. coli* TOP10 and verified using automated DNA sequencing (Sangon, Shanghai, China).

## Bioinformatic analyses

The open reading frame (ORF) and encoded amino acid sequence of CHI were analyzed using ORF Finder (https://www.ncbi.nlm.nih.gov/orffinder/) followed by translation of the nucleotide sequence using DNAMAN software. Domain structure was analyzed at the National Center for Biotechnology Information (NCBI) (https://blast.ncbi.nlm.nih.gov/ Blast.cgi) and the three-dimensional structure of CHI was predicted using SWISS-MODEL (https://swissmodel.expasy.org/) (*Jiang et al., 2017*). A phylogenetic neighbor-joining tree was constructed using MEGA 5.0 software, and ClustalW was used for sequence alignment (*Liu et al., 2018*; *Noby et al., 2018*).

## Expression and purification of chitinase

For the expression of recombinant CHI, pET-22b (+)-CHI was transformed into *E. coli* Rosetta-gami B (DE3) cells. Cells were cultured at 37 °C in liquid LB medium containing 100 μg mL$^{-1}$ ampicillin. Expression was induced by the addition of 0.5 mM isopropyl thio-β-D-galactoside (IPTG) when the $OD_{600}$ reached 0.6–1.0, after which cells were incubated at 37 °C for a further 8 h. Protein expression was checked by sodium dodecyl sulfate polyacrylamide gel electrophoresis (SDS-PAGE) (10% separation gel, 4% concentrated gel). The cell pellet was collected by centrifugation at 8,000 rpm for 10 min and stored at 4 °C. The freshly-prepared cell pellet was resuspended in lysis buffer (0.1 M Tris-HCl, pH 8.1, containing 0.3 M NaCl), then lysed on ice using an ultrasonic cell disrupter (JY92-IIN; Ningbo Scientz Biotechnology Co., LTD., Ningbo, China) followed by centrifugation (8,000 rpm, 15 min, 4 °C) to remove cell debris and unbroken cells. The supernatant, comprising crude protein extracts containing CHI, was immediately applied to a 5 mL Ni-NTA affinity chromatography column pre-equilibrated with binding buffer (0.1 M Tris-HCl, pH 8.1, 0.3 M NaCl and 10 mM imidazole) at 4 °C. The column was washed with 50 mL of wash buffers (0.1 M Tris-HCl, pH 8.1, 0.3 M NaCl and 20 mM imidazole, then 0.1 M Tris-HCl, pH 8.1, 0.3 M NaCl and 60 mM imidazole). Chitinase was eluted with 40 mL buffer containing 0.1 M Tris-HCl (pH 8.1), 0.3 M NaCl, and 150 mM and 250 mM imidazole, respectively. Purity of the recombinant protein was analyzed using 10% SDS-PAGE with Coomassie Brilliant Blue R-250 staining. The molecular mass of the purified enzyme was determined by comparison with protein

molecular weight markers (Product #: 26616, Thermo Scientific, Beijing, China). The eluted fractions which contained target protein were concentrated and exchanged into a 50 mM phosphate buffer, pH 7.0, using 30 kDa centrifugal filter units (Millipore, Beijing, China). Protein concentrations were determined using the Bradford method (*Bradford, 1976*), and enzyme preparations were stored at 4 °C until use.

## Biochemical characterization of chitinase

The optimum temperature for CHI activity was determined by incubating enzyme preparations with colloidal chitin at various temperatures between 25 °C and 80 °C in 50 mM phosphate buffer at pH 7.0. Thermostability was tested by preincubating the enzyme at various temperatures from 25 °C to 70 °C without substrate for 60 or 90 min. The optimum pH was determined by carrying out the reaction at pH 2.0–11.0 with 0.2 M phosphate (pH 2.0–8.0) or 0.2 M glycine-NaOH (pH 8.0–11.0) buffers. Chitinase was preincubated at pH 2.0–11.0 in the appropriate buffer at 4 °C for 5 h prior to the activity assay. The residual enzyme activity was tested at the optimum pH. The effects of metal ions and chemical reagents ($Ca^{2+}$, $Mg^{2+}$, $Zn^{2+}$, $Na^+$, $K^+$, $Al^{3+}$, $NH_4^+$, $Co^{2+}$, $Ba^{2+}$, $Cu^{2+}$, $Fe^{3+}$, $Mn^{2+}$, $Ag^+$, $Cd^{2+}$, Tris, SDS), ethylenediaminetetraacetic acid (EDTA), and urea) on CHI activity at final concentrations of 10 and 50 mM were also investigated, as were the effects of organic solvents and reagents (methanol, ethanol, isopropanol, glycerin, isoamyl alcohol, Tween 80, acetic acid, chloroform, dimethylsulfoxide (DMSO), and β-mercaptoethanol) at final concentrations of 10% and 20% (v/v). The effect of NaCl on CHI activity was determined by incubating the enzyme with NaCl at final concentrations of 0.5%, 1.5%, 3%, 4.5%, 6%, 7.5%, 9%, or 12.5% at 4 °C for 1 h. Residual CHI activity was measured in standard assay conditions. Appropriate controls were used.

## Substrate specificity

The specificity of recombinant CHI was tested toward various substrates such as colloidal chitin, chitin powder, chitosan oligosaccharide, carboxymethyl cellulose sodium salt (CMC-Na) and carboxymethyl chitosan by adding the compounds to the assay mixture individually at a final concentration of 0.5% (w/v). Relative enzyme activity was then determined in standard conditions with the highest activity being taken as 100%.

## Product analysis

The major products of colloidal chitin hydrolysis by CHI was identified using electrospray ionization-mass spectrometry (ESI-MS; AB Sciex, Framingham, Ma, USA) analysis. The reaction mixture containing 100 μL colloidal chitin (1%) and 0.25 mg CHI in 50 mM 100 μL phosphate buffer (pH 7.0) was incubated at 45 °C for 0.5 h, 1 h, 6 h and 14 h. After incubation, the reaction mixture was boiled for 5 min and centrifuged at 8,000 rpm for 10 min at 4 °C. The supernatant was analyzed by ESI-MS.

# RESULTS

## Expression and purification of chitinase

The gene encoding chitinase was successfully isolated from *P. chitinolyticus* strain UMBR 0002 and recombinant CHI was heterologously expressed in *E. coli* Rosetta-gami B (DE3)

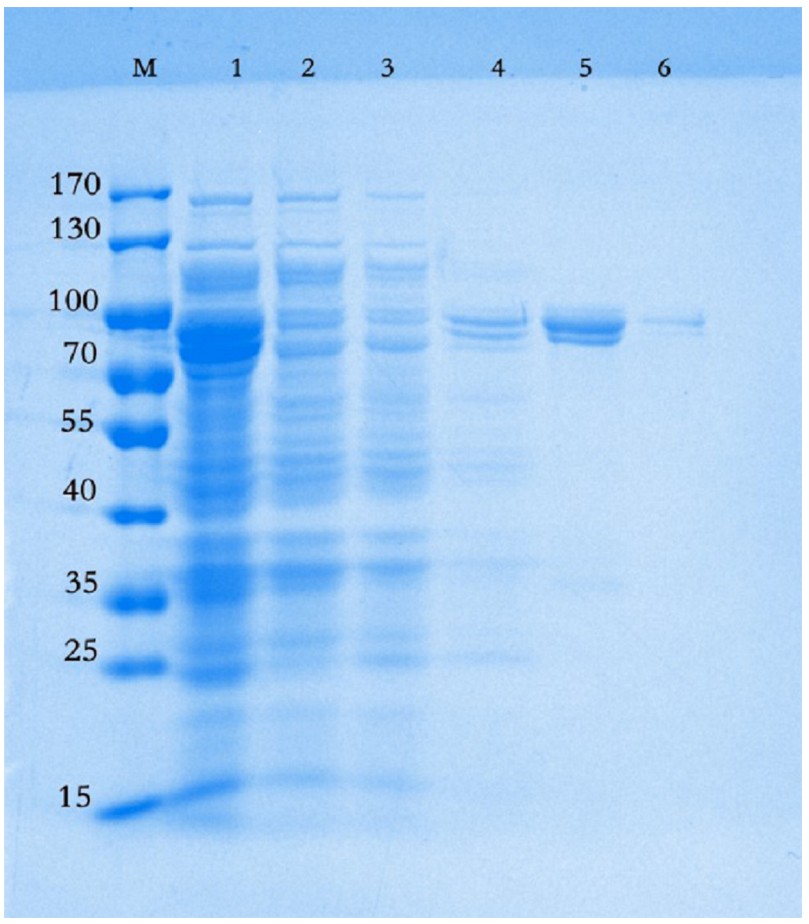

**Figure 1 Representative gel of purification steps of chitinase from *Paenibacillus chitinolyticus* strain UMBR 0002.** M, Protein molecular weight markers; 1, Crude extract of *Escherichia coli* expressing chitinase purified from *Paenibacillus chitinolyticus* strain UMBR 0002 (CHI); 2, Unbound protein; 3, Protein eluted with buffer containing 20 mM imidazole; 4, Protein eluted with buffer containing 60 mM imidazole; 5, Protein eluted with buffer containing 150 mM imidazole; 6, Protein eluted with buffer containing 250 mM imidazole. The gel was stained with Coomassie Brilliant Blue R-250.

**Table 1 Chitinase purification.**

| Purification step | Volume (mL) | Protein concentration (mg mL$^{-1}$) | Enzyme activity (mU mL$^{-1}$) | Specific activity (mU mg$^{-1}$) | Total activity (mU) | Purification (fold) | Yield% (total activity) |
|---|---|---|---|---|---|---|---|
| Crude enzyme solution | 20 | 4.16 | 233.80 | 56.19 | 4,676.10 | 1 | 100 |
| Ni-NTA affinity chromatography | 2 | 2.25 | 1,688.12 | 750.64 | 3,376.25 | 13.36 | 72.20 |

and purified. The molecular weight was estimated to be between 70 and 100 kDa by SDS-PAGE (Fig. 1), which is consistent with the expected weight of approximately 80 kDa according to the translated nucleotide sequence. In our study, recombinant CHI was purified with 13.36-fold enrichment and a recovery yield of 72.20%. The purified enzyme had a specific activity of 750.64 mU mg$^{-1}$ (Table 1).

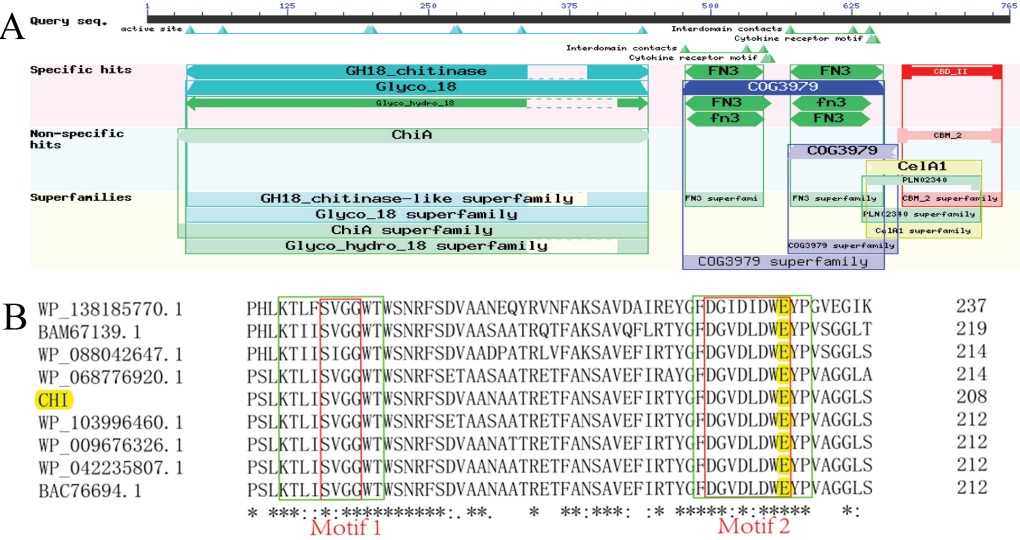

**Figure 2 Domain architectures and multiple sequence alignment of chitinase from *Paenibacillus chitinolyticus* strain UMBR 0002.** (A) Domain structures of chitinase from *Paenibacillus chitinolyticus* strain UMBR 0002. Protein sequence analysis was carried out using BLASTP on NCBI. The locations of the following domains are indicated: GH18_chitinase (aa: 36-445; Accession, cd06548); fibronectin type III (FN3; aa:477-547; Accession, smart00060); fibronectin type III (FN3; aa: 571-653; Accession, cd00063), CBD_II (CBD_II; aa:670-785, Accession, smart00637). (B) Deduced amino acid sequence alignment of open reading frames of chitinase with chitinases from other species. Green boxes indicate the two conserved motifs KxxxxxGGW and FDGxDLDWEYP; red boxes are typical conserved modules of the glycoside hydrolase family 18, SxGG and DxxDxDxE. The yellow E indicates glutamic acid.                                   

## Bioinformatic analysis

The gene encoding CHI included a complete ORF encoding a protein of 765 amino acids. Domain structure analysis revealed CHI to contain a chitin-binding domain (CBD) and two fibronectin-type-III domains (FN3). The enzyme was identified to belong to glycoside hydrolase family 18 (Fig. 2A). Sequence alignment using BLASTP software (NCBI) revealed that CHI exhibits 98.26% sequence identity and 97% sequence coverage with chitinase from *P. chitinolyticus* (WP_042235807.1), followed by chitinase Chi80 from *P. ehimensis* (BAC76694.1) (97.58% identity and 97% sequence coverage), chitinase from *Paenibacillus* sp.UNC499MF (WP_103996460.1) (95.49% identity and 95% sequence coverage), chitinase from *Paenibacillus* sp. FJAT-26967 (WP_068776920.1) (79.59% identity and 94% sequence coverage), and chitinase from *Bacillus* sp. EAC (WP_088042647.1) (62.62% identity and 94% sequence coverage). Multiple sequence alignment showed that, although the coding amino acid sequences of different chitinases vary greatly, the key amino acids within the catalytic domain are highly conserved. In this study, two conserved motifs with the sequences KxxxxxGGW and FDGxDLDWEYP were identified in CHI (Fig. 2B). Within these conserved sequences, CHI has two conserved modules typical of the glycoside hydrolase family 18, SxGG and DxxDxDxE, which are responsible for substrate binding and catalytic degradation, respectively. Glutamic acid (E) represents a general acid/base (catalytic acid/base) for the protonation of

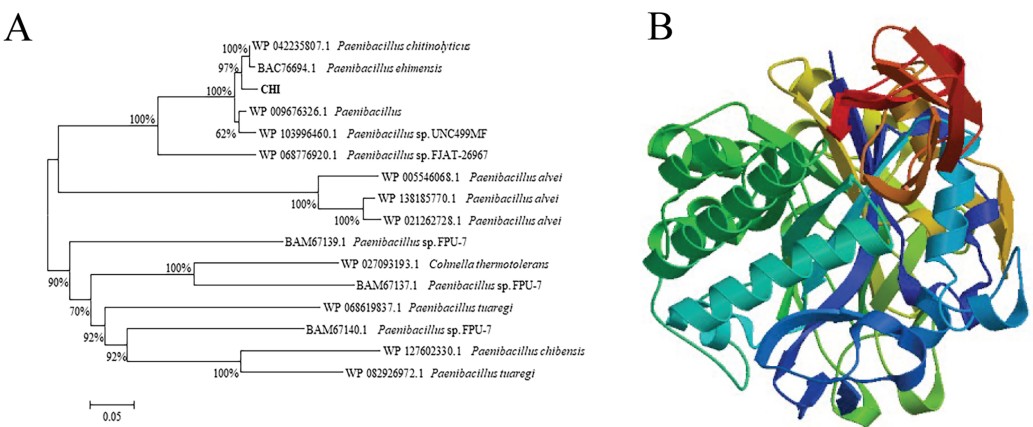

**Figure 3** Phylogenetic analysis and homology model of chitinase from *Paenibacillus chitinolyticus* strain UMBR 0002. (A) Amino acid sequence similarity tree based on sequences of proteins from glycoside hydrolase family 18 (*n* = 16). The phylogenetic tree was constructed using the neighbor-joining option of ClustalW software. GenBank accession number for chitinase gene from *Paenibacillus chitinolyticus* strain UMBR 0002: MN121846. (B) Three-dimensional (3D) structure model of chitinase from *Paenibacillus chitinolyticus* strain UMBR 0002.               

oxygen atoms from sugar molecules. Phylogenetic analysis revealed that CHI shares close evolutionary relationships with chitinases from *P. ehimensis* and *P. chitinolyticus* (Fig. 3A). Comparisons using the Protein Data Bank (PDB) revealed the homology of CHI and Chitinase ChiA 74 from *B. thuringiensis* to be 62.40%, with high matching accuracy. The results of homologous modeling are illustrated in Fig. 3B.

## Effect of temperature and pH on enzyme activity and stability

The determined temperature-activity profile of CHI is shown in Fig. 4A. Chitinase activity increased gradually with increasing temperature from 25 to 45 °C; maximal activity was observed at 45 °C. Relative enzyme activity of >90% was retained at 50 °C. Above 50 °C, the activity of CHI decreased markedly. The thermal stability of CHI is illustrated in Fig. 4B. The enzyme was generally stable at 25–45 °C for both periods of incubation.

The effect of pH on the activity of CHI is shown in Fig. 4C. The highest activity was observed within the pH range 4.0–8.0, with maximal activity recorded at pH 5.0 (relative activities at pH 4.0 and 7.0 were 53.16% and 71.02% respectively). Activity at pH 8.0 was higher in glycine-NaOH than phosphate buffer. The effect of pH on CHI stability is shown in Fig. 4D. The optimum pH for stability of this chitinase was 8.0, but the enzyme was very stable within the pH range 4.0–10.0 (with relative activity of >70% observed after incubation at 4 °C for 5 h).

## Effect of chemical reagents, metal ions, and solvents on enzyme activity and stability

The influence of metal ions and chemical reagents on the activity of purified CHI is shown in Fig. 5. The presence of $Mn^{2+}$, $Ca^{2+}$ and urea significantly increased the activity of CHI. We found that the activity of CHI was not significantly affected by $Mg^{2+}$, $Na^+$, $K^+$, or

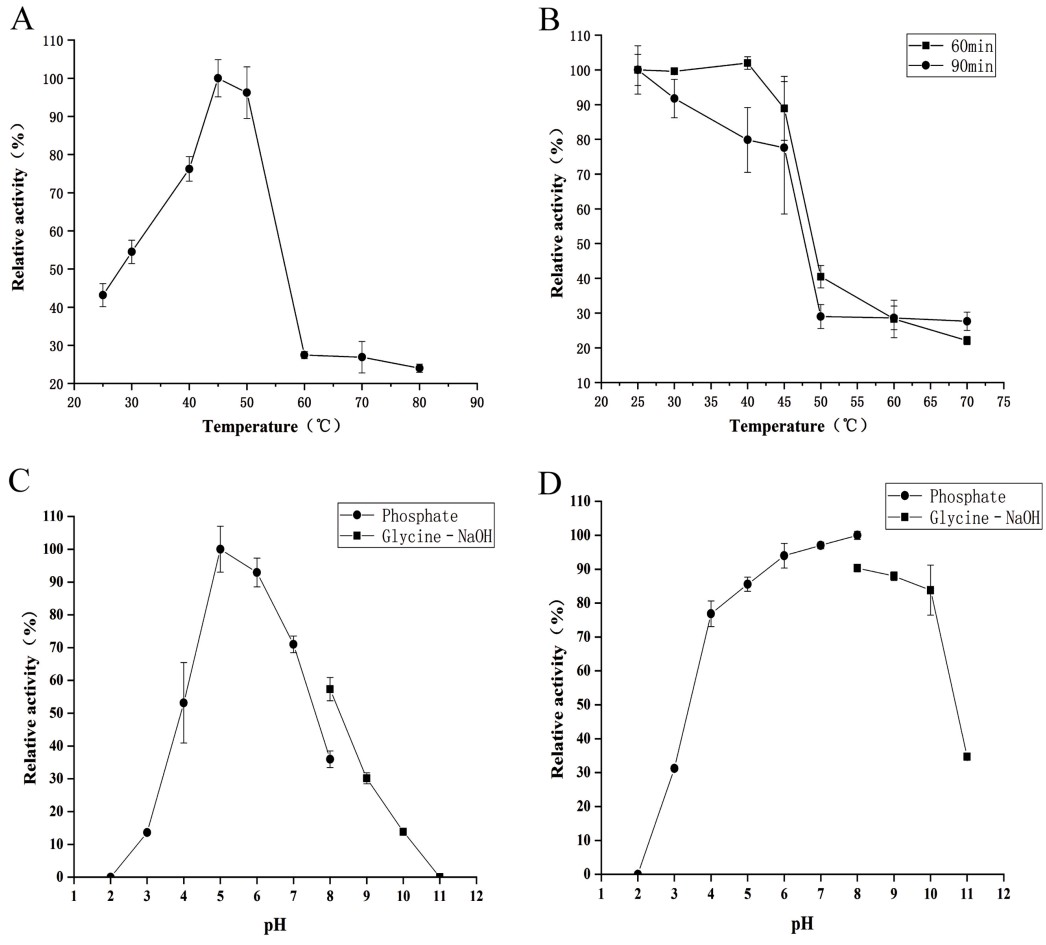

**Figure 4 Effects of temperature and pH on the activity and stability of chitinase from *Paenibacillus chitinolyticus* strain UMBR 0002.** (A) The optimum temperature for activity of chitinase from *Paenibacillus chitinolyticus* strain UMBR 0002 (CHI) toward colloidal chitin at pH 7.0. The activity of the enzyme at 45 °C was defined as 100%. (B) Thermostability of CHI at pH 7.0. Activity of the non-heated enzyme was defined as 100%. (C) Activity of CHI toward colloidal chitin at various pH levels at 45 °C. The activity of the enzyme at pH 5.0 was defined as 100%. (D) Stability of CHI after incubation at various pH levels at 4 °C for 5 h. The activity of the enzyme after incubation at pH 8.0 (in phosphate buffer) was defined as 100%. Each data point represents the mean of three independent experiments, standard errors are shown.

$NH_4^+$ in the present study. When mixed with 10 mM $Co^{2+}$, Tris, or $Cd^{2+}$, the residual activity of CHI was high, but decreased moderately when these reagents were present at 50 mM. Relatively high activity (>100%) was maintained in the presence of 10 mM $Ba^{2+}$, although the activity dropped sharply when the concentration of $Ba^{2+}$ was increased to 50 mM. We found CHI activity to be strongly inhibited at both tested concentrations of $Zn^{2+}$, $Al^{3+}$, $Ag^+$ and SDS (relative activity of <60% in each case). However, the presence of $Cu^{2+}$ and $Fe^{3+}$ obviously inhibited the activity of CHI. We also found that 10 mM EDTA enhanced activity, while 50 mM EDTA significantly reduced activity.

The influence of various organic solvents on the stability of purified CHI is shown in Fig. 6; the activity obviously increased in the presence of Tween 80. In addition,

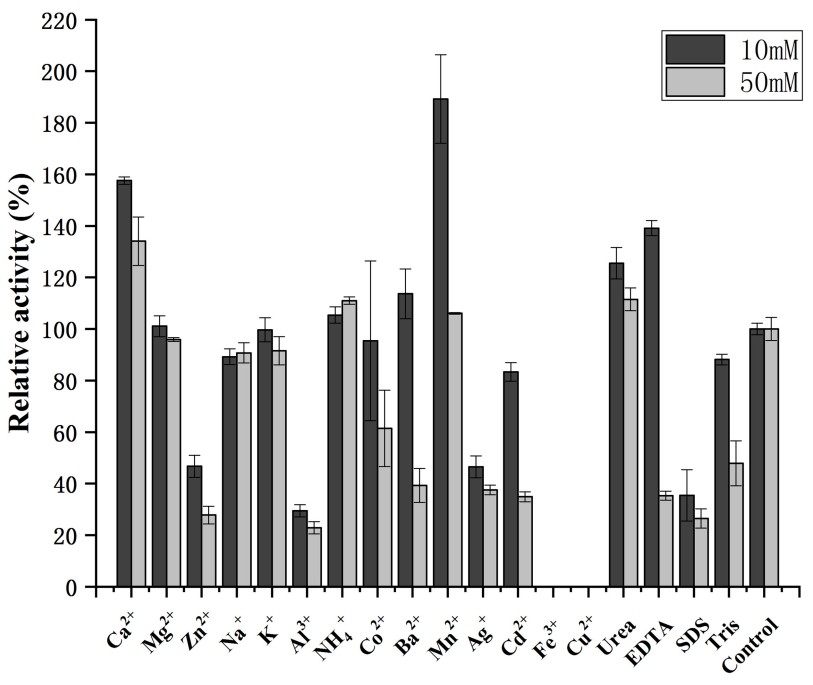

**Figure 5 Effects of chemical reagents and metal ions on activity of chitinase from *Paenibacillus chitinolyticus* strain UMBR 0002.** Untreated enzyme was used as the control, and the activity of this preparation defined as 100%. Experiments were carried out in triplicate and standard errors are shown. Abbreviations: EDTA, ethylenediaminetetraacetic acid; SDS, sodium dodecyl sulfate.

CHI activity was moderately increased (relative activity >100%) in the presence of 10% chloroform, although activity decreased sharply when 20% chloroform was used. Most of the organic solvents tested in the present study; including methanol, ethanol, isopropanol, glycerin and DMSO; caused CHI activity to decrease somewhat when used at 10% final concentration, while 20% organic solvent markedly suppressed activity. However, β-mercaptoethanol, isoamyl alcohol and acetic acid strongly inhibited CHI activity (by >50%) at both concentrations.

## Effect of NaCl concentration on enzyme activity

As Fig. 7 shows, the optimum salt concentration was found to be 1.5% NaCl. At higher concentrations, chitinase activity decreased, reaching <60% relative activity at 12.5% NaCl.

## Substrate specificity

The specific activities of purified recombinant CHI toward different substrates are shown in Table 2. The highest specific activity was toward colloidal chitin (567.20 mU mg$^{-1}$). The next highest activity was observed when chitin powder was used as a substrate (288.69 mU mg$^{-1}$); in this case, the relative activity was 50.90% ± 3.04% compared with colloidal chitin (100%). No activity was detected toward chitosan oligosaccharide (DDA > 90%), carboxymethyl chitosan (DDA ≥ 90%) or CMC-Na.

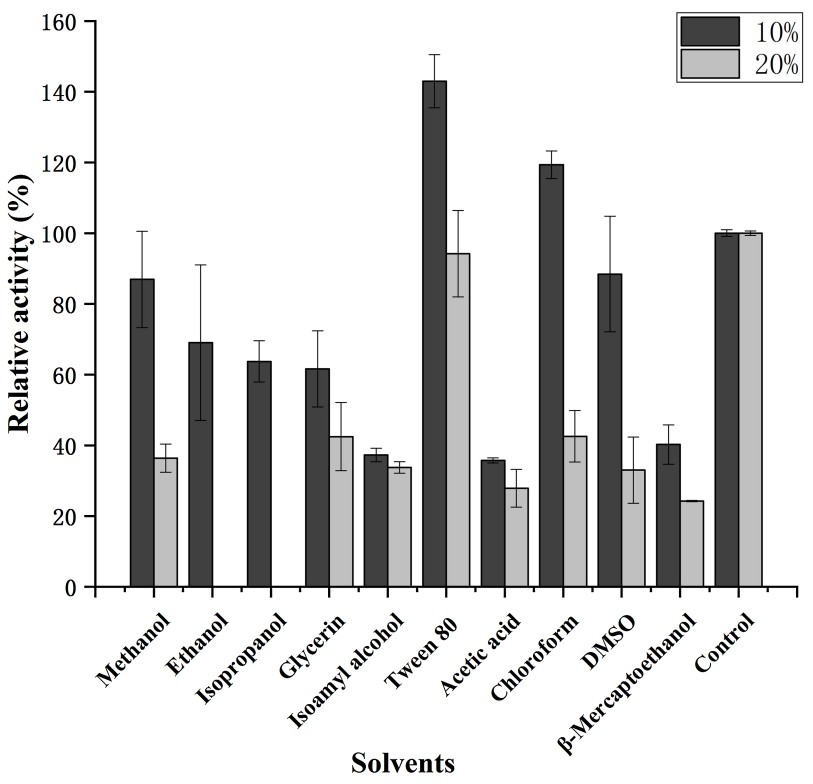

**Figure 6 Effect of solvents on the activity of chitinase from *Paenibacillus chitinolyticus* strain UMBR 0002.** Untreated enzyme was used as the control (defined as 100% activity). Experiments were carried out in triplicate and standard errors are shown. Abbreviations: DMSO, dimethylsulfoxide.

## Mass spectrometry analysis of the hydrolysis product

We used ESI-MS in positive ion mode to analyze the major products of colloidal chitin hydrolysis by purified CHI and the degradation mode of chitinase was inferred. As shown in Figs. 8 and 9, peaks with *m/z* of 244/260 (GlcNAc), 447/463 (GlcNAc)$_2$ and 650 (GlcNAc)$_3$ were attributed to sodium, potassium, or hydrogen adducts of oligosaccharides with different degrees of polymerization. ESI-MS analyses showed the major hydrolysis product of colloidal chitin to be GlcNAc, with some (GlcNAc)$_2$ produced (Fig. 9).

## DISCUSSION

In this study, we cloned and heterologously expressed the CHI gene from marine-derived *P. chitinolyticus* strain UMBR 0002, and functionally characterized the CHI properties and products, with the aim of providing an increased understanding of the purified CHI activity in *P. chitinolyticus* strain UMBR 0002, and benefiting to the extensive studies of marine-derived bacteria as novel sources of chitinases for the green degradation of chitin waste. The majority of previous studies on *P. chitinolyticus* have focused on the wild bacterium (*Song et al., 2012*; *Mihajlovski et al., 2017*). However, cloning and expression of CHI from *P. chitinolyticus* strain UMBR 0002 are rarely performed to study furtherly specific enzymatic properties. Therefore, in-depth studies of the characteristics of

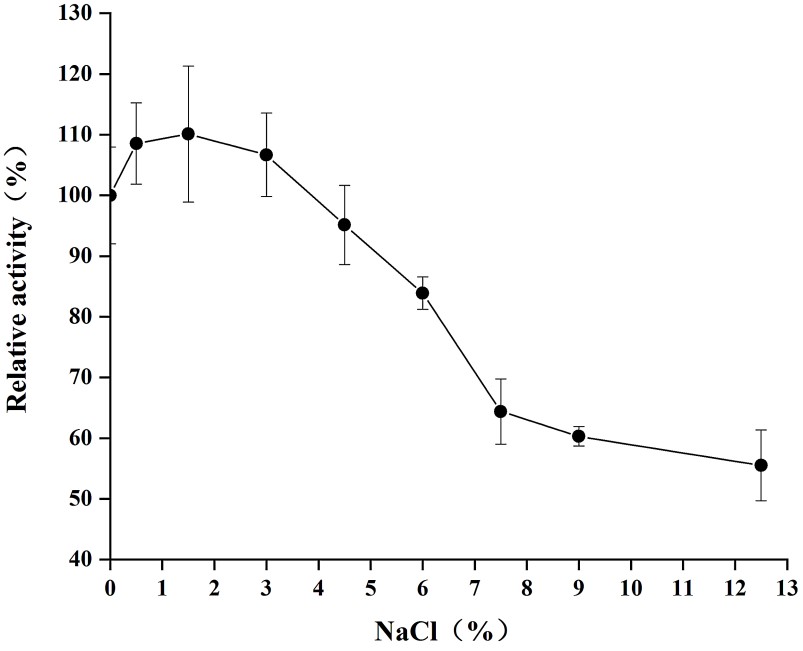

**Figure 7 Effect of salt concentration on the activity of chitinase from *Paenibacillus chitinolyticus* strain UMBR 0002.** Untreated enzyme was used as the control and defined as 100% relative activity. Experiments were carried out in triplicate and standard errors are shown.

**Table 2 Substrate specificity of chitinase from *Paenibacillus chitinolyticus* strain UMBR 0002.**

| Substrate | Specific activity[a] (mU mg$^{-1}$) | Relative activity[a,b] (%) |
|---|---|---|
| Colloidal chitin | 567.20 ± 34.48 | 100 ± 6.08 |
| Chitin powder | 288.69 ± 17.22 | 50.90 ± 3.04 |
| Chitosan oligosaccharide | 0 | 0 |
| Carboxymethyl chitosan | 0 | 0 |
| CMC-Na | 0 | 0 |

Notes:
[a] Experiments were conducted three times and standard errors are reported.
[b] Relative enzyme activity calculation using colloidal chitin as the reference value (100%). Abbreviations: CMC-Na, carboxymethyl cellulose sodium salt.

*P. chitinolyticus* strain UMBR 0002 chitinase may enable the development of procedures for the efficient degradation of chitin waste.

Most chitinases from *Paenibacillus* species are reported to be in the range of 38–153 kDa (*Fu et al., 2014*); the molecular masse of CHI (approximately 80 kDa) in the present is close to chitinases from other *Paenibacillus* species such as *Paenibacillus* sp. FPU-7 (61, 78, 82, 87,97, 122 and 153 kDa) (*Itoh et al., 2013*), *P. ehimensis* MA2012 (>100, 100, 72, 65, 60, 50, 37 and 35 kDa) (*Seo et al., 2016*) and *P. barengoltzii* (about 74.6 kDa) (*Fu et al., 2014*). Interestingly, the zymogram of mature CHI exhibited about two single bands surrounded by clear zones (Fig. 1), similar to results of analysis of chitinase obtained from the Antarctic psychrotolerant bacterium, *Vibrio* sp. strain Fi:7, which produced protein bands of 80 and 82 kDa on the zymogram (*Bendt et al., 2001*). The reason for the two

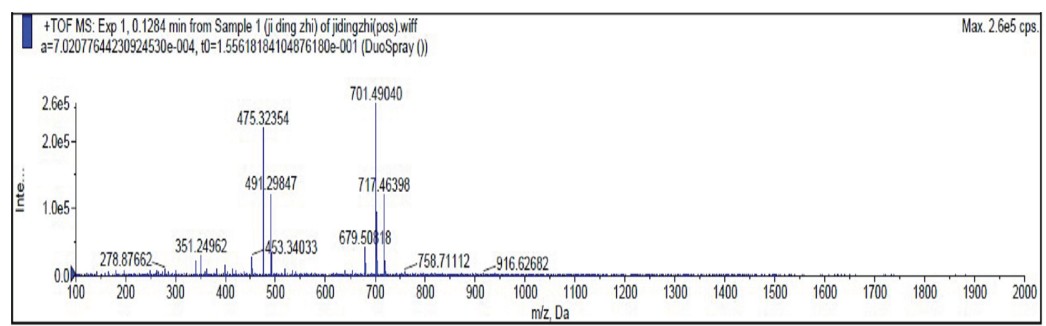

**Figure 8 Electrospray ionization-mass spectrometry spectrum of colloidal chitin.**

bands observed in the present study remains unclear and requires further study. One possible explanation could be that CHI exists in two isoforms (*Bendt et al., 2001*), as has been reported for two chitinases from *Streptomyces* sp. (*Okazaki et al., 1995*).

CHI exhibited its optimum temperature at 45 °C, which was lower than that reported for chitinase from *P. thermoaerophilus* strain TC22-2b (60 °C) (*Ueda & Kurosawa, 2015*), but higher than that of chitinase from *P. pasadenensis* NCIM 5434 (37 °C) (*Loni et al., 2014*). CHI showed high stability, retaining more than 70% activity at 25–45 °C (maximum of 90 min), which is similar to the results reported for chitinase from *S. marcescens* B4A (*Emruzi et al., 2018*). The charge distribution of substrate and enzyme molecules can affect substrate binding and catalysis (*Bouacem et al., 2018*). The optimum pH for chitinases from *Hydrogenophilus hirschii* strain KB-DZ44 (*Bouacem et al., 2018*), *Paenibacillus* sp. D1 (*Singh & Chhatpar, 2011*) and *P. illinoisensis* strain KJA-424 (*Jung et al., 2005*) has been reported to be around 5.0, in line with results of the present study; however, chitinases from *Sulfolobus tokodaii* (*Staufenberger, Imhoff & Labes, 2012*), *P. thermoaerophilus* strain TC22-2b (*Ueda & Kurosawa, 2015*), *S. antarcticus* KOPRI 21702 (*Park et al., 2009*), and *P. pasadenensis* NCIM 5434 (*Loni et al., 2014*) are reported to exhibit maximum activity at pH 2.5, 4.0, 7.6 and 10.0, respectively. The present results indicate that CHI was highly stable at pH 4.0–10.0 for 5 h at 4 °C. This is similar to chitinase from *P. thermoaerophilus* strain TC22-2b, which was reported to be stable at pH 4.0–10.0 (*Ueda & Kurosawa, 2015*), but a wider range than chitinase from *P. barengoltzii* (pH 4.0–9.0) (*Fu et al., 2014*). The broad pH range and high pH stability suggest that CHI may be a good candidate for industrial and commercial applications (*Ueda & Kurosawa, 2015*). Metal ions play an important role in biocatalysts by stabilizing substrate-enzyme complexes (*Andreini et al., 2008*). Enzyme activity was enhanced by almost two fold in the presence of 10 mM $Mn^{2+}$ compared with the control (no metal ions). This is similar to the results of a previous study chitinase from *Melghiribacillus thermohalophilus* strain Nari2AT (*Mohamed et al., 2019*). We also found that $Ca^{2+}$ (10 and 50 mM) and urea caused enzyme activity to increase. It is well known that metal ions act as cofactors for enzyme activity and contribute to the maintenance of enzyme structure by acting as ion or salt bridges (*Kamran et al., 2015*). Industrial extraction of chitin usually involves the dissolve of minerals such as calcium carbonate from shrimp and crab shells

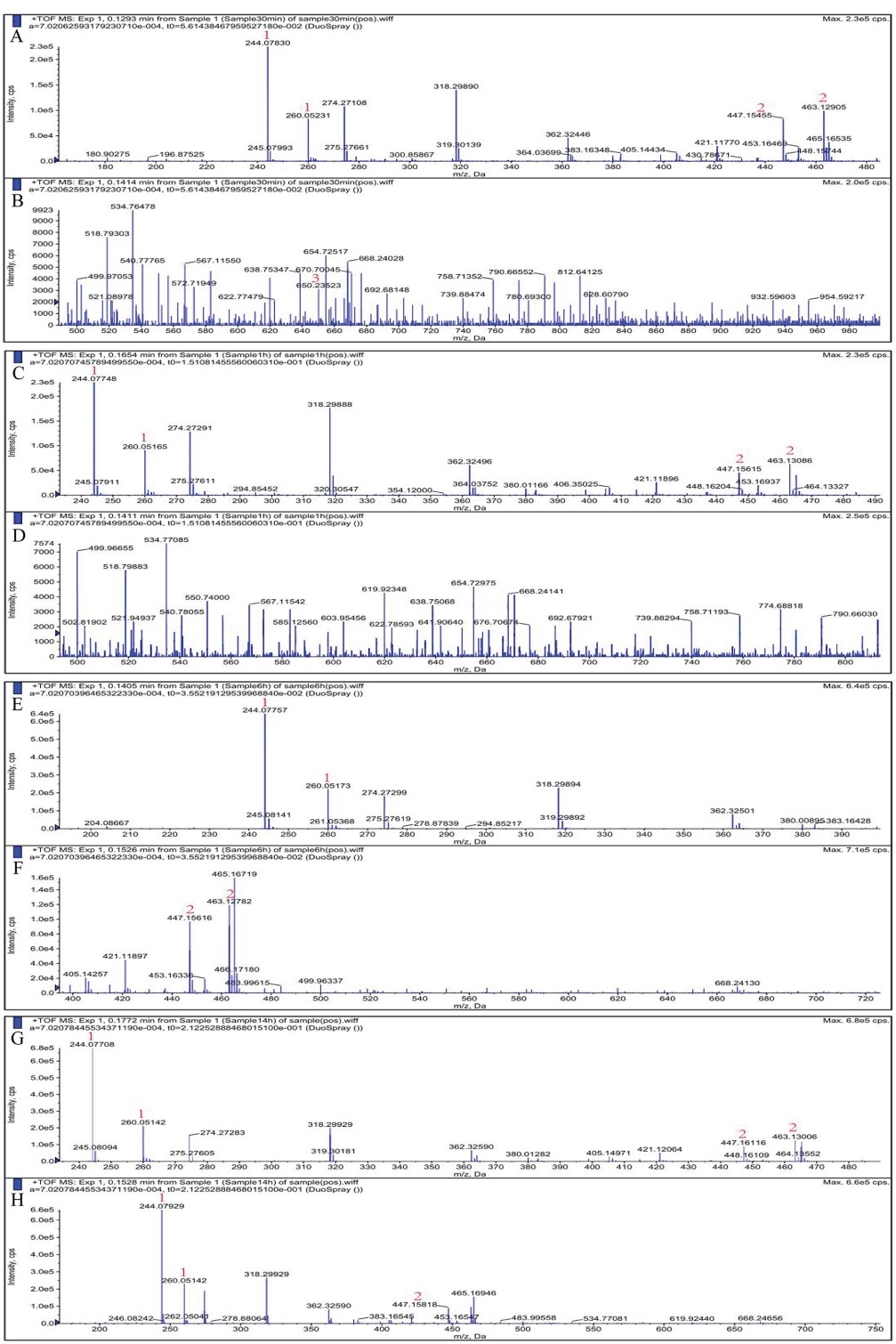

**Figure 9 Electrospray ionization-mass spectrometry spectra of products of hydrolysis of colloidal chitin by chitinase from *Paenibacillus chitinolyticus* strain UMBR 0002.** Products of enzymatic hydrolysis were analyzed after incubation for (A and B): 0.5 h, (C and D): 1 h, (E and F): 6 h, and (G and H): 14 h. Peak 1: (GlcNAc); 2: (GlcNAc)$_2$; 3: (GlcNAc)$_3$. Abbreviations: GlcNAc, N-acetylglucosamine.

and releases $Ca^{2+}$, followed by further processing to obtain valuable chitin derivatives. Therefore, CHI can offer advantages in industrial application. The chitinase analyzed in this study might resemble a chitinase from *P. pasadenensis* NCIM 5434, which was reported to be increased by $Ca^{2+}$ at 100 mM concentration (*Loni et al., 2014*). This is in contrast with chitinase from *Glaciozyma antarctic*a PI12 (*Ramli et al., 2011*), which has been reported to be inhibited by 10 mM $Ca^{2+}$. In the present study, CHI activity was completely inhibited by $Cu^{2+}$ and $Fe^{3+}$; heavy metal ions may denature enzymes by destroying the tertiary structure of the protein, thus inactivating the molecules. This result is in line with observations of chitinase from *P. pasadenensis* NCIM 5434, where enzymatic activity was completely inhibited by $Cu^{2+}$ (*Loni et al., 2014*). It has been suggested that $Cu^{2+}$ catalyzes the autooxidation of cysteine, resulting in the formation of intramolecular disulfide bonds or sulfonic acid, which may inhibit the activity of chitinase (*Deng et al., 2019*). The influence of EDTA on purified CHI is consistent with chitinase from *Chitinolyticbacter meiyuanensis* SYBC-H1 (*Zhang et al., 2018*), whose activity was found to be enhanced by 10 mM EDTA. As EDTA is a metal ion chelator EDTA, the effect on enzyme activity indicates that CHI is not dependent on metal ions for enzymatic activity. The reported effects of metallic ions on chitinases are diverse (*Mohamed et al., 2019*) and future studies investigating the mechanism underlying the effects of metals on chitinase activity are important. Organic solvents are used to solubilize hydrophobic substrates in enzymatic assays. The activity of purified CHI was also obviously increased by the presence of Tween 80, similar to chitinase from *S. thermodiastaticus* HF 3-3 (*Take et al., 2018*). This indicates that Tween 80 reduces the activation energy of the reaction, promoting the enzymatic reaction. The effects of many organic solvents in the study on purified CHI activity are similar to those relating to chitinases produced by *Streptomyces* sp. (*Karthik, Binod & Pandey, 2015*) and *Aeromonas hydrophila* SBK1 (*Halder et al., 2016*). The stability of CHI in low concentrations of most organic solvents indicates that hydrophobic interactions play an important role in enzymatic activity, and may make it possible to use this enzyme in practical industrial applications (*Mohamed et al., 2019*). Salt-tolerant chitinases play important roles in industrial operations (*Karthik, Binod & Pandey, 2015*). Activity of CHI could be increased by addition of NaCl (0–3%) in the reaction (Fig. 7). Moreover, more than 90% residual activity was recorded at NaCl concentrations of up to 4.5%. The concentration of NaCl in seawater is about 3%; thus, CHI may have utility for the degradation of chitin waste from sea food and waste from marine and coastal environments.

To examine the substrate specificity of CHI, various substrates were selected as shown in Table 2. Our results indicated that CHI preferred to colloidal chitin than chitin powder. Which indicates that CHI can degrade colloidal chitin most readily of the tested substrates. This degradation may be followed by further hydrolysis. The DDA of chitosan directly influences its biological activities. A greater DDA reflects a larger number of positively charged amine groups in acidic conditions (*He et al., 2016*). In the catalytic process of GH18 chitinase, the acid that protonates the glycosidic linkage is a conserved glutamate and the nucleophile is the oxygen of the N-acetyl group on the −1 sugar and the reaction proceeds via an oxazolinium ion intermediate. Due to this unusual enzymatic

mechanism, a GlcNAc residue is required in the −1 subsite for catalytic cleavage to occur (*Hartl, Zach & Seidl-seiboth, 2012*). This explains the lack of activity that we detected toward chitosan oligosaccharide (DDA > 90%) and carboxymethyl chitosan (DDA ≥ 90%). We did not detect any activity toward CMC-Na, similar to chitinase from *Aeromonas* sp. No. 16 which cannot degrade CMC (*Huang, Chen & Su, 1996*).

For a deeper understanding of the catalytic mechanism of CHI, the major products of colloidal chitin hydrolyzed by purified CHI (Figs. 8 and 9) were examined by ESI-MS. Colloidal chitin can become deacetylated under alkaline conditions; therefore, the presence of GlcNAc in Fig. 9 is unsurprising. After 0.5 h (Figs. 9A and 9B), considerable hydrolysis of colloidal chitin was observed, producing GlcNAc and $(GlcNAc)_2$ as the main products, and $(GlcNAc)_3$ as a minor product. This also indicates that CHI does not have endochitinase activity. At 1 h (Figs. 9C and 9D), the minor product, $(GlcNAc)_3$, was not detected, and GlcNAc and $(GlcNAc)_2$ were present in seemingly equal amounts. From 6 to 14 h (Figs. 9E and 9H), the abundance of the dominant product, GlcNAc, continued to increase, while the change of $(GlcNAc)_2$ was not obvious in the reaction system. These results suggest that, with sufficient degradation time, CHI can hydrolyze substrates to produce (GlcNAc) as the major product and $(GlcNAc)_2$ as a minor product. The products of colloidal chitin hydrolysis are consistent with the hydrolysates reported to be formed by exochitinases derived from *B. thuringiensis* (*Juárez-Hernández et al., 2019*), which has the highest similarity (62.40%) to CHI from homology modeling. Therefore, CHI can be considered to be an exochitinase with β-N-acetylglucosaminidase activity. The hydrolysis mechanism and processing of CHI are similar to those of PbChi74 from *P. barengoltzii*, which possesses two chitinolytic activities (exochitinase and N-acetyl-β-glucosaminidase activities) (*Fu et al., 2014*). However, the hydrolysis mechanism of CHI is different to that of an exochitinase (ChiC) from *Pseudoalteromonas* sp. DL-6, which was only found to have exochitinase activity requiring a prolonged reaction time to hydrolyze colloidal chitin (*Wang et al., 2016*). An exochitinase of *S. speibonae* TKU048 has been demonstrated to possess N-acetyl-β-glucosaminidase activity only (*Tran et al., 2019*), while a novel chitinase (CmChi1) from *C. meiyuanensis* SYBC-H1 is reported to possess both exo- and endochitinase abilities with poor N-acetyl-β-glucosaminidase activity (*Zhang et al., 2018*). Biological production is a valuable "green" method for GlcNAc production; therefore, the hydrolysis activity of CHI may be beneficial for industrial production of GlcNAc from chitin materials. Nevertheless, the specific catalytic mechanism underlying the hydrolysis of chitin by CHI needs to be further studied.

## CONCLUSIONS

In summary, we report the successful cloning and purification of a chitinase from *P. chitinolyticus* strain UMBR 0002 using an *E. coli* Rosetta-gami B (DE3) expression system. The enzyme, which was named CHI, had a molecular mass of approximately 80 kDa and optimum catalytic activity toward colloidal chitin at 45 °C and pH 5.0, respectively. Significant substrate specificity toward colloidal chitin was identified and ESI-MS analyses showed the major hydrolysis product of colloidal chitin to be GlcNAc,

with some $(GlcNAc)_2$ produced. Products were obtained in a relatively short time, indicating that CHI is an exochitinase. High activity and stability were observed over wide temperature, pH and (NaCl) ranges; furthermore, the presence of $Ca^{2+}$ and Tween 80 significantly increased chitinase activity. These properties make this enzyme a promising candidate for green industrial conversion of chitinous waste to chitin oligomers. Taken together, CHI could be exploited for biological and environmental applications in the future. The results presented here deepen our understanding of the *Paenibacillus* species and could provide insight into other enzymes from this strain.

## ACKNOWLEDGEMENTS

We thank Amy Phillips, PhD, from Liwen Bianji, Edanz Editing China for editing the English text of a draft of this manuscript.

### Funding

This research was funded by the National Natural Science Foundation of China (81960164, 31660022 and 31660005); National Natural Science Foundation of Guangxi province, China (2018GXNSFAA281113); Science and Technology Major Project of Guangxi (AA18242026); Specific Research Project of Guangxi for Research Bases and Talents (AD18126005, AD18281066); Xiangsihu Young Scholars Innovative Research Team of Guangxi University for Nationalities (201704); The Scientific Research Project for Introducing High-level Talents of Guangxi University for Nationalities (2018KJQD1, 2018KJQD17); Basic Ability Improvement Project of Guangxi University for Young and Middle-Aged Teachers (2019KY0192, 2017KY0165) and the Innovation Project of Guangxi Graduate Education (gxun-chxzs2018061). The funders had no role in study design, data collection and analysis, decision to publish, or preparation of the manuscript.

### Grant Disclosures

The following grant information was disclosed by the authors:
National Natural Science Foundation of China: 81960164, 31660022 and 31660005.
National Natural Science Foundation of Guangxi province, China: 2018GXNSFAA281113.
Science and Technology Major Project of Guangxi: AA18242026.
Specific Research Project of Guangxi for Research Bases and Talents: AD18126005 and AD18281066.
Xiangsihu Young Scholars Innovative Research Team of Guangxi University for Nationalities: 201704.
The Scientific Research Project for Introducing High-level Talents of Guangxi University for Nationalities: 2018KJQD1 and 2018KJQD17.
Basic Ability Improvement Project of Guangxi University for Young and Middle-Aged Teachers: 2019KY0192 and 2017KY0165.
Innovation Project of Guangxi Graduate Education: gxun-chxzs2018061.

## Competing Interests

The authors declare that they have no competing interests.

## Author Contributions

- Cong Liu conceived and designed the experiments, performed the experiments, analyzed the data, prepared figures and/or tables, authored or reviewed drafts of the paper, and approved the final draft.
- Naikun Shen conceived and designed the experiments, analyzed the data, authored or reviewed drafts of the paper, and approved the final draft.
- Jiafa Wu conceived and designed the experiments, analyzed the data, authored or reviewed drafts of the paper, and approved the final draft.
- Mingguo Jiang conceived and designed the experiments, analyzed the data, authored or reviewed drafts of the paper, and approved the final draft.
- Songbiao Shi analyzed the data, authored or reviewed drafts of the paper, offered guidance and supervised the project, and approved the final draft.
- Jinzi Wang analyzed the data, authored or reviewed drafts of the paper, offered guidance and supervised the project, and approved the final draft.
- Yanye Wei conceived and designed the experiments, prepared figures and/or tables, offered guidance and supervised the project, and approved the final draft.
- Lifang Yang conceived and designed the experiments, analyzed the data, authored or reviewed drafts of the paper, and approved the final draft.

## DNA Deposition

The following information was supplied regarding the deposition of DNA sequences:
The CHI sequences are available at GenBank: MN121846.

## Data Availability

The raw data are available in the Supplemental Files.

## Supplemental Information

Supplemental information for this article can be found online at http://dx.doi.org/10.7717/peerj.8964#supplemental-information.

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
