# Peer review of "Cloning, expression and characterization of a chitinase from Paenibacillus chitinolyticus strain UMBR 0002"

_PeerJ, doi:10.7717/peerj.8964_

## Round 0.1 · original submission · Major Revisions

Please consider using a professional editorial service since all reviewers concur in the need for extensive editorial corrections. Please attend to all reviewers in a detailed rebuttal letter.

Reviewer 1 ·

Basic reporting

no comment

Experimental design

no comment

Validity of the findings

no comment

Additional comments

The work described in the manuscript entitle “Cloning, expression and characterization of a chitinase from Paenibacillus chitinolyticus strain”, although not very new, deserves to be published in this Journal. Chitinases have great utility in different areas of Biotechnology, so the characterization and study of new enzymes with this activity would already constitute an interesting piece of work for biotechnologists. However, the following general and particular aspects must be improved.

General aspects
The main problem I see in this work is that many data that readers need to know to say if the work is novel, or not, have been omitted. For example, the characterized gene (the protagonist of this story) is very similar to different genes from organisms of the same genera, among them, from a different strain of the same species (Fig.2), and nothing is said about the % identity (sequence coverage!) of the corresponding encoded proteins, nor if there were any characterized chitinase from this microbial species. Nor is anything said about how many potential chitinolytic genes have been found in the genome of the Paenibacillus spp, and P. chitinolyticus, or about sizes of the potential chitinases, which can be deduced from the genome already sequenced, or if had any of them been characterized from a functional point of view, etc.
In general, the full text should be reviewed and corrected.

Some particular aspects
1. Ln 55; Chitin can be cleaved into chitosan!, it is beaks into chitooligosaccharides!
2. Ln 57: GlcNAc or Glc NAc?, unify throughout the manuscript. Take care of the writing because this abbreviation is described in Ln 133 and is used repeatedly before!
3. Ln 68; three structural families based on their structural motives not on their sequences!
4. Ln 103; vector (no vectors) pET-22b(+) and it is not enough to say that it is already in a laboratory, indicate provider
5. Ln 117; What kind of commercial chitin, in shrimp flakes? Is chitin powder already soluble in water?, and if not in what? and form colloids?
6. Ln 118 ….acid mM?
7. Ln 122; how the colloid mass was calculated to indicate concentration in g / L
8. Ln 133; the definition of OD is indicated later in the text, in ln 166!
9. Lns 139-142; genera of organisms are abbreviated from the 2nd time they are named!.
10. Ln 182; which protein was used as a control pattern?
11. Ln 204; Carboxymethyl cellulose/chitosan: suppliers, how are they prepared?
12. Ln 217; gene was isolated from ..not cloned from..
13. Ln 218; CHI was heterologously expressed in …
14. Lns 221-224, any explication for the 2 bands?, Both have chitinolytic activity ?, could have done a triptic digestion and MALDI of both proteins to see if one is a product of hydrolysis or a less glycosylated protein variant?
15. Lns 230-232; give data on the analyzed sequence, indicate found domains, position them by adding between what and what amino acids are the domains, etc ... Are there any sequence / domain of binding to Ca, Mg, etc. since they are then used and the activity of the enzyme changes?
16. Ln 236, Fig 2. Phylogenetic tree: to give % identity and sequence coverage!!
17. Ln268-280; Very repetitive, the data that are already shown in Fig 4 are described! and nothing is said about Mn, which is the one that produces the greatest effects! Sort and summarize all this.
18. Ln 301..industrial applications, to give examples/references…
19. Lns 317-322; mass of some non-totally acetylated chitooligosaccharide is also not detected in Fi8?, for example, that of 405.144 could be due to GlcNAc-GlcN, and if this is so, this should also be commented and discussed.
20. Table 2, relative percent column, it says nothing, repetitive with the part of the specific activity; few data to justify a table. All this could be indicated in the text..
21. Bibliography has to be reviewed and unified! now, before indicating the pages (, or :), the names of the journals abbreviated or complete, volume number in italics or not….

Reviewer 2 ·

Basic reporting

It is difficult for reviewer to understand the contents of this manuscript, in particular introduction section. Authors should improve this manuscript throughout using language editing service.

Experimental design

Authors stated that One unit of chitinase activity was defined as the amount of enzyme required to produce 1 micromol of GlcNAc from colloidal chitin per min per liter of solution at 45 °C. Does 3,5-dinitrosalicylic acid (DNS) react with only monomer, GlcNAc? Does DNS react with reducing ends of oligosaccharide or of polysaccharide? Is it correct ‘per min per liter’, that is U/L? Why did enzyme reaction time require 60 minute at 45 degrees C? Authors should describe in a careful manner. For example, in legend of Figure 1 reviewer could not find ‘Crude extract’ and ‘250 mM imidazole’ in materials and method section.

Validity of the findings

Authors stated that ‘This is the first report that cloning, heterologous expression and purification of a chitinase (CHI) from Paenibacillus chitinolyticus strain UMBR 0002, and the results indicate that CHI is a good candidate enzyme for the green degradation of chitinous waste’. However, about CHI authors should emphasize what is superior to other reported chitinases, what is similar to other chitinases, and should discuss why CHI is different from other reported enzymes and is similar to them, because some chitinases from Paenibacillus sp. were well studied and a chitinase is one of enzymes that are more resistant to pH and temperature.

Additional comments

Line69 glycoside hydrolase families 18, 19 and 20. Not glycoside hydrolase chitinase families
Line85 should refer a reference
Line88 Are enzyme activity (4.16 and 3.65 U/mL) high compared with other chitinases?
Line103 should state company etc. not stored.
Line 221 should show the zymogram. Could authors analyze both N-terminal sequences of two bands?
Line 232 Does CHI possess CBD at N-terminal side of catalytic domain?
Line 236 should compare with other reported chitinases that are well studied on phylogenetic analysis and show their amino acid sequences. Title of Figure 2, Amino acid sequence similarity tree, is strange. Reviewer could not access MN121846 on the National Center for Biotechnology Information.
Line 242 novel Streptomyces strain (Hoster, Schmitz & Daniel, 2005). Is it novel?
Line 265 purified CHI activity
Line 267-274 should explain why Mn2+ and Ca2+ increase CHI activity.
Line 280 should explain why authors think that Cu2+ and Fe3+ may destroy the 3D structure
Line 313 should state acetylated value or deacetylated value of chitosan. All glucosamine (GlcN)?
Line 320-322 It is known exo-typed chitinases could produce GlcNAc, GlcNAc2, and GlcNAc3. Therefore, the statement that CHI is endo-typed chitinase is questionable. If CHI has characteristics of endo-typed chitinase on primary structure compared with endo-chitinases, could discuss them.

Reviewer 3 ·

Basic reporting

The manuscript needs extended revision from a native English speaker. This is a very good scientific work with good data and novelty. Due my native language is not English; no corrections on my behalf have been submitted.

Experimental design

No comments. The approach is pertinent and meaningful. Description of methods is good.

Validity of the findings

No comments

Additional comments

This is a very good scientific work; it only needs editing in English. I suggest the authors to get help in English to present their valuable work and submit again a reedited version to the journal.

---

## Round 0.2 · Minor Revisions

To continue the review process, we need the GenBank accession number of the exact sequence overexpressed, from where the primers were designed. We also need the information on the SDS*-PAGE molecular weight markers, brand, and technical info.

---

## Round 0.3 · Minor Revisions

The authors have addressed the technical issues requested. However, it is important to solve whether they have deposited in GenBank the nucleotide sequence used for expression since it contains 6 histidines at the C-terminus, if that is the case, request an amendment to GenBank curators.

---

## Round 0.4 · accepted · Accept

The manuscript has improved over the review rounds and it is now accepted at PeerJ.